# Translation of Partially Paired Images with Generative Adversarial Networks

Sachin Chhabra*, Yaoxin Zhuo*, Riti Paul*, Javad Sohankar†, Ji Luo†, Shan Li†, Wendy Lee†,
Yi Su†, Teresa Wu* and Baoxin Li*
*SCAI, Arizona State University, Tempe, USA
†Banner Alzheimer's Institute, Phoenix, USA
Email: *{sachin.chhabra, yzhuo6, rpaul12, Teresa.Wu, Baoxin.Li}@asu.edu,
†{mohammadjavad.sohankaresfahani, ji.luo, shan.li, wendy.lee, yi.su}@bannerhealth.com

*Abstract*—The integration of paired medical MRI (Magnetic Resonance Imaging) and PET (Positron Emission Tomography) images holds considerable significance in clinical evaluations and offers a richer source of clinical insights. However, acquiring paired MRI-PET images poses challenges due to the various practical constraints. To address this, MRI-PET translation emerges as a valuable approach, enabling professionals to obtain complementary information from one modality and enhance decision-making using only single-modality images. Existing approaches predominantly rely on either using paired MRI-PET images for training or treating the entire dataset as unpaired. In this study, we introduce PaPaGAN, an innovative end-to-end Partially Paired Generative Adversarial Network specifically tailored for partially paired images. In a practical setting, where a mix of paired and unpaired data is available, PaPaGAN leverages the unpaired data to learn a mapping function capable of generating a noisy intermediate image. To refine this intermediate image and address the inconsistencies during the unpaired translation process, PaPaGAN employs a secondary image translation module. This module is specifically trained using the paired data, which provides a consistent mapping from source to target domain images. By effectively harnessing both paired and unpaired MRI-PET images, our method significantly enhances translation capabilities, facilitating precise image translation and elevating image quality for the target modality. Our quantitative and qualitative medical image translation experiments on two public datasets, ADNI and OASIS, demonstrate the superiority of PaPaGAN over alternative image translation methods.

*Index Terms*—Medical Image Translation, PaPaGAN, Partially Paired Translation.

## I. INTRODUCTION

Recent advancements in medical image-to-image translation have revolutionized clinical practices, streamlining workflows and opening new avenues for diagnostic and treatment planning [15], [20], [23]. These methods are very effective in generating realistic medical images across diverse modalities, democratizing access to medical imaging, and potentially revolutionizing patient care and research. An exemplary application lies in the translation between Magnetic Resonance Imaging (MRI) and Positron Emission Tomography (PET). While MRI provides intricate details of soft tissues,

This work was supported in part by the following grants: National Institutes of Health Grant RF1AG073424, National Institutes of Health Grant P30AG072980, and Arizona Department of Health Services Grant CTR057001. Any opinions expressed in this material are those of the authors and do not necessarily reflect the views of funding agencies.

aiding in the assessment of brain structure and pathology, PET highlights metabolic activities crucial for diseases like Alzheimer's. The synergistic combination of both modalities offers invaluable insights for accurate diagnosis and treatment decisions. However, obtaining both MRI and PET scans for every patient poses challenges due to cost constraints, equipment availability, and concerns over radiation exposure. In such cases, image-to-image translation models can be employed to enhance decision-making by providing complementary information of the missing modality.

Generative adversarial networks (GANs) [9] is a popular image-generation technique and serve as the cornerstone for several deep learning-based image-generation techniques [17], [19], including image translation [1], [4], [12], [30]. These image translation methods can broadly be classified into paired and unpaired approaches. Paired medical image translation relies on access to matched MRI-PET image pairs for model training, whereas unpaired methods operate on MRI and PET images without requiring pairwise correspondences.

Pix2pix [12] stands out as a leading technique for image-to-image translation, particularly suited for paired data scenarios. It employs an image generator and an image discriminator. The discriminator's objective is to differentiate between the original target image and the images synthesized by the generator. The generator tries to outsmart the discriminator and, in turn, produces realistic images that closely mimic the target domain. By harnessing the power of paired images, Pix2pix excels in enhancing pixel-to-pixel level similarity, thereby preserving fine-grained details during translation. Other techniques for paired data follow the principle with some extensions. However, the availability of paired data is scarce, particularly in the medical domain, where data collection can be arduous and resource-intensive.

For image translation using unpaired data, which is often easier to collect, CycleGAN [30] is the most popular technique. It employs two generator-discriminator pairs, one focusing on the source domain while the other tackles the target domain. One generator is tasked with converting images from the source domain to the target domain, while the other performs the inverse transformation. Similarly, one discriminator focuses on the source domain, and the other tackles the target domain. The generators are trained to fool the

discriminator along with a cycle consistency loss that helps to preserve the main content of the image. The cycle consistency loss is an unreliable method for preserving the content as an image can be translated to something entirely different and then reconstructed as the original when translated back. Most unpaired data training approaches are based on CycleGAN with limited enhancements.

Our method, **PaPaGAN**, is specifically designed to handle partially paired data, where some data is paired while the rest remains unpaired. This is a more realistic setting as it is always easy to get unpaired data, which can be combined with paired data to create a partially paired dataset. Traditional paired image translation methods overlook unpaired data, missing out on potential enhancements. Conversely, unpaired approaches neglect the valuable information available in paired data, potentially yielding suboptimal translations. PaPaGAN bridges this gap by leveraging both paired and unpaired data effectively.

Our approach employs two generator-discriminator pairs trained with a mix of paired and unpaired data (treated as unpaired data only) and cycle consistency loss to translate images into the target domain. The output from these generators is then fed into a secondary generator-discriminator pair trained exclusively on paired data. Therefore, the input image passes through two image generator modules to achieve the final target image. The first generator creates an image in the target domain but emphasizes encoding the source properties during its reconstruction due to the cycle consistency loss. However, this loss can be ineffective when significant differences exist between domains. To address this, we introduce a second image generator that takes the source image and the output of the first generator to produce a more accurate version.

PaPaGAN is a bi-directional translation method that simultaneously trains networks for both translation directions. This bi-directionality allows us to use the same discriminator for translating paired data in both directions, effectively regularizing the networks. In this work, our contributions can be summarized as follows:

- PaPaGAN is designed to work effectively with partially paired datasets, which include both paired and unpaired data. This setting is more realistic and practical in many real-world applications where unpaired data is more readily available than paired data.
- To address the limitations of cycle consistency loss, especially when there are significant differences between domains, PaPaGAN introduces a second generator to produce an enhanced translation.
- Comprehensive evaluations on benchmark datasets such as ADNI [26] and OASIS [13] demonstrate that PaPaGAN outperforms unpaired and paired baselines. Experiments show that the partially paired data-based frameworks provide advantages that neither setting (paired/unpaired) can achieve alone. This highlights the innovative aspect of PaPaGAN in integrating multiple levels of supervision for superior performance.

## II. RELATED WORK

While PET is a relatively new modality compared to non-invasive MRI, its application in diagnosis is rising. However, obtaining both MRI and PET scans for each patient is often impractical due to the high costs, limited availability of PET scanners, and concerns over radiation exposure. To mitigate these issues, recent research has focused on generating PET data from existing MRI or CT scans. For example, Li et al. [14] utilized a 3D-CNN to learn the non-linear mapping between paired MRI-PET scans from the ADNI dataset. Gao et al. [8] introduced RIED-Net, which aims to produce higher-quality PET images from MRI data (also from the ADNI dataset). These techniques, however, often result in smoothed images that lose detail because they use L1/L2 optimization methods. Notably, the task of MRI-to-PET translation is related to, but distinct from, MRI super-resolution (e.g., [6]), where the output is higher-resolution MRI images, and domain adaptation (e.g., [2], [3]), where classification is typically the objective.

Generative Adversarial Networks (GANs) have recently shown significant success in image generation and restoration tasks. Due to their ability to produce high-quality synthetic images, GANs have been integrated into various multi-modal or cross-modal medical imaging tasks. Pan et al. [18] proposed a 3D conditional GAN (cGAN) framework to model bi-directional mappings between MRI and PET scans for Alzheimer's Disease (AD) diagnosis. Yaakub et al. [28] designed a 3D GAN with residual connections to learn the mapping from MRI to PET for evaluating patients with focal epilepsy. Sikka et al. [22] employed a method that enhances both global structural integrity and local detail fidelity in synthetic PET images using a multi-path architecture. Other significant architectures utilizing paired data for training include Pix2Pix [12], CoCaGAN [11], and EA-GAN [29].

Despite their success, these frameworks depend heavily on paired cross-modal data, which is costly and labor-intensive to collect and annotate. Several unpaired image translation frameworks [4], [5], [30] have been developed recently to tackle the challenge of paired data scarcity. For instance, Zhu et al. [30] introduced CycleGAN, which incorporates a cycle consistency loss specifically designed for unpaired medical image generation. StarGAN [4], [5] uses a discriminator that distinguishes between fake and real images and performs auxiliary domain classification. Another notable method for unpaired image-to-image translation is the UNsupervised Image-to-image Translation (UNIT) [16], which is based on a Variational Autoencoder (VAE)-GAN framework.

In our work, we address a more common scenario where data often exist in a partially paired form; some data is paired while most remain unpaired. Instead of relying solely on cycle consistency loss [30], we propose to improve or correct a noisy approximation of the translated image using paired data mapping. The proposed method helps to preserve structural details from the source and incorporate target domain style adaptation for enhanced image quality.

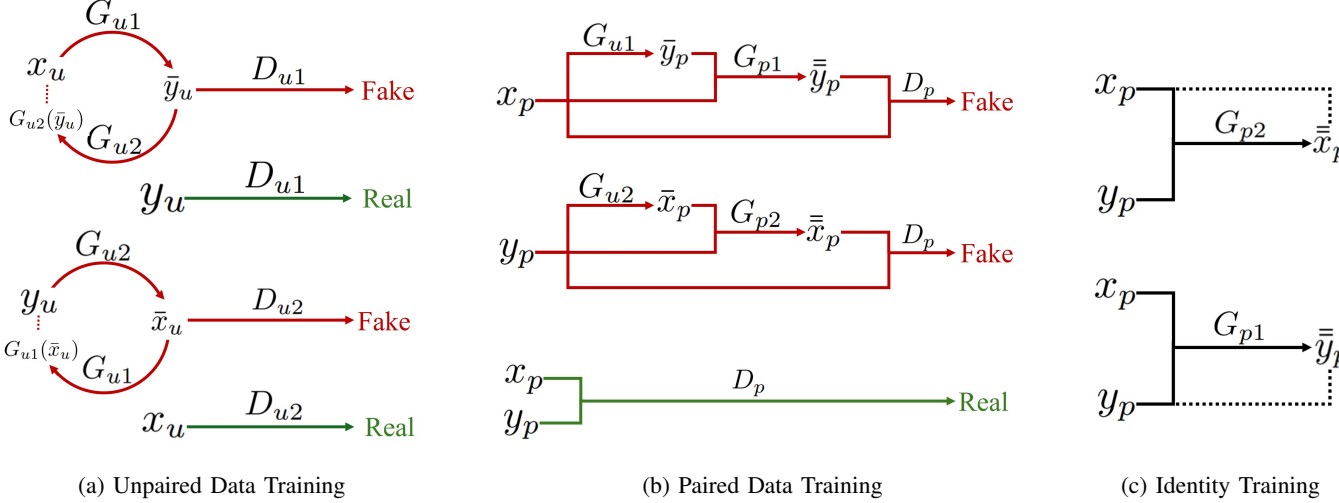

(a) Unpaired Data Training      (b) Paired Data Training      (c) Identity Training

Fig. 1: Model Diagram of our approach. (a) shows the cycle consistency and adversarial training with unpaired data. (b) demonstrates usage training of image generators with paired data. (c) shows identity loss, which promotes paired generators to use correct inputs from unpaired generators. Dotted lines denote Identity loss (L1-loss) between two images.

## III. METHOD

### A. Problem formulation

Consider a dataset $P \in \{x_p, y_p\}^{N_p}$ comprising $N_p$ pairs of source and target domain images, where $x_p$ represents a source image and $y_p$ denotes its corresponding image from the target domain. Additionally, let $U \in \{x_u\}^{N_x}, \{y_u\}^{N_y}$ represent an unpaired dataset containing $N_x$ source images and $N_y$ target images. The objective of PaPaGAN is to learn an optimal image translation mapping between the source and target domains by utilizing both the paired data $P$ and the unpaired data $U$.

### B. Unpaired Data Training

Partially Paired data compromises of both paired $P$ and unpaired $U$ data. However, paired data can also be treated as unpaired data. Hence, we first combine $P$ paired and $U$ unpaired datasets to form a larger unpaired dataset $U'$ of size $N_p + N_u$. By doing this, we effectively increase the size of available unpaired data. More data helps networks approximate their domains better and also regularize the networks.

The training with unpaired data compromises two pairs of generator and discrimination, namely $G_{u1}$, $G_{u2}$, $D_{u1}$, and $D_{u2}$, which are trained using generative adversarial network framework [9]. The generators $G_{u1}$ take $x$ as input and outputs $\bar{y} = G_{u1}(x)$ and similarly we get $\bar{x} = G_{u2}(y)$. We train the discriminators $D_{u1}$ and $D_{u2}$ to differentiate real vs generated images using,

$$\mathcal{L}_{ud} = -\mathbb{E}_{x,y \sim U'}[(1 - D_{u1}(y))^2 + D_{u1}(G_{u1}(x))^2 + (1 - D_{u2}(x))^2 + D_{u2}(G_{u2}(y))^2] \quad (1)$$

The generators are tasked to fool the discriminators, which in turn helps them to generate realistic images as,

$$\mathcal{L}_{ug}^{gan} = -\mathbb{E}_{x,y \sim U'}[(1 - D_{u1}(G_{u1}(x)))^2 + (1 - D_{u2}(G_{u2}(y)))^2] \quad (2)$$

We also train the image generators with cycle-consistency loss [30] to ensure network outputs do not refrain too much from the original input,

$$\mathcal{L}_{ud}^{cc} = -\mathbb{E}_{x,y \sim U'}[(x - G_{u2}(G_{u1}(x)))^2 + (y - G_{u1}(G_{u2}(y)))^2] \quad (3)$$

The two training loss functions of image generators are balanced using a hyper-parameter $\lambda_u = 10$. The final training loss for the image generator is,

$$\mathcal{L}_{ug} = \mathcal{L}_{ug}^{gan} + \lambda_u \mathcal{L}_{ug}^{cc} \quad (4)$$

### C. Paired Data Training

Unpaired data training uses cycle consistency loss to ensure the content does not change much during the translation. However, in cases with higher domain differences, such as cross-modal medical imaging, the effectiveness of cycle consistency loss tends to deteriorate. In our case, MRI and PET modalities contain significant domain differences. In such cases, cycle consistency forces the network to encode source properties for reconstruction rather than preserving visual content.

To combat this problem, we refine the generated target image using another set of generators $G_{p1}$ and $G_{p2}$. The generators $G_{p1}$ and $G_{p2}$ are specialized for task of correcting mistakes of $G_{u1}$ and $G_{u2}$ respectively. They take the output from the unpaired generators $G_{u1}$ and $G_{u2}$ along with their inputs $x$ and $y$ and learn to generate the correct outputs using paired data $\bar{\bar{y}} = G_{p1}(x; G_{u1}(x))$ and $\bar{\bar{x}} = G_{p2}(G_{u2}(y); y)$ respectively.

Unlike traditional GANs, we train the generators $G_{p1}$ and $G_{p2}$ using a single discriminator $D_p$ for both domains instead of two discriminators. The discriminator $D_p$ is a conditional discriminator that takes both source and target images as input.

To provide multiple inputs in the generators and discriminator, we fuse the source and the target domain images along the channel dimension, doubling the number of channels in their inputs. This strategy enables $G_{p1}$, $G_{p2}$ and $D_p$ to learn the structural similarities inherent in paired source and target domain images. By doing so, the discriminator is better equipped to ensure that the generator preserves the essential content and structural details during the translation, resulting in more accurate and realistic outputs. It also enables us to use the same discriminator for both domains, effectively increasing its training data. We train the $D_p$ to differentiate real vs fake pairs of both the domains as,

$$\mathcal{L}_{pd} = -\mathbb{E}_{(x,y)\sim P}[(1 - D_p(x;y))^2 +$$
$$\frac{1}{2}(D_p(x;G_{p1}(x;G_{u1}(x))))^2 + \frac{1}{2}(D_p(G_{p2}(G_{u2}(y);y);y))^2] \quad (5)$$

The $\frac{1}{2}$ weightage is given to each fake generated image pair to avoid any bias towards the fake class due to the imbalance. The generators are tasked to fool the discriminator,

$$\mathcal{L}_{pg}^{gan} = -\mathbb{E}_{(x,y)\sim P}[(1 - D_p(x;G_{p1}(x;G_{u1}(x))))^2 +$$
$$(1 - D_p(G_{p2}(G_{u2}(y));y);y)^2] \quad (6)$$

Integrating the GAN losses with conventional metrics such as $L_1$ and $L_2$ distances has proven beneficial for convergence [12], [30]. This accelerates convergence rates and further encourages generated outputs to closely resemble the ground truth. Thus, we incorporate the $L_1$ distance into the training objective of the generators as,

$$\mathcal{L}_{pg}^{dist} = \mathbb{E}_{(x,y)\sim P}||y - G_{p1}(x;G_{u1}(x))||_1 +$$
$$||x - G_{p2}(G_{u2}(y);y)||_1 \quad (7)$$

Similar to unpaired data training, we balance the generator losses using $\lambda_p = 10$ hyperparameter.

### D. Identity Training

The primary goal is to ensure that the final output images are as accurate as possible. To achieve this, the paired generators refine and correct the output of the unpaired generators. However, if the output of unpaired generators is optimal, we want the paired generators to recognize and retain optimal outputs from the unpaired generators. Hence, we add an identity loss for the paired generators as,

$$\mathcal{L}_{pg}^{identity} = \mathbb{E}_{(x,y)\sim P}||y - G_{p1}(x;y)||_1 +$$
$$||x - G_{p2}(x;y)||_1 \quad (8)$$

The only difference between 7 and 8 is that we use the ground truth target image as part of the input to the paired generators. By incorporating the ground truth target image into the input during training, the paired generators learn to recognize correct outputs. This training method ensures that

paired generators can effectively utilize the correct inputs without altering them unnecessarily. It also augments their training data and enhances the overall performance.

### E. Training procedure

We follow the standard procedure of training discriminators and generators iteratively. First, we train the unpaired discriminators $D_{u1}$ and $D_{u1}$ on unpaired data and the paired discriminator $D_p$ on paired data, respectively. Following discriminator training, we train the combined generators framework $G_{u1}$, $G_{u2}$, $G_{p1}$, and $G_{p2}$ of paired and unpaired data with the combined loss functions,

$$\mathcal{L}_g = \mathcal{L}_{ug}^{gan} + \lambda_u \mathcal{L}_{ug}^{dist} + \mathcal{L}_{pg}^{gan} + \lambda_p \mathcal{L}_{pg}^{dist} + \lambda_i \mathcal{L}_{pg}^{identity} \quad (9)$$

$\lambda_u$, $\lambda_p$, and $\lambda_i$ are loss-weighing hyperparameters and are set to 10, 10, and 1, respectively. We generate the final output target image as $\bar{\bar{y}} = G_{p1}(x;G_{u1}(x))$ and $\bar{\bar{y}} = G_{p2}(G_{u2}(y);y)$ for domains using the trained generators and inputs $x$ and $y$ respectively. Our approach is modeled in Figure 1.

## IV. EXPERIMENTS

### A. Settings

*1) Dataset:* To assess the effectiveness of PaPaGAN, we conducted experiments on two prominent public MRI-PET datasets: ADNI (Alzheimer's Disease Neuroimaging Initiative) and OASIS-3 (Open Access Series of Imaging Studies-3). For a comprehensive evaluation, we performed image translation experiments in both directions: from MRI→PET and vice versa. ADNI is a multisite study aimed at enhancing clinical trials for Alzheimer's Disease (AD) prevention and treatment. From ADNI, we collected 2,717 paired 3D MRI-PET samples after excluding any corrupted or missing data. Additionally, we gathered an additional 2,174 MRI samples lacking paired PET counterparts. OASIS-3, collected by Washington University, is a third-generation database obtained with a waiver of informed consent. After filtering out corrupted or missing data, we obtained 770 3D paired MRI-PET and 2,071 extra MRI samples, respectively. The PET data for both of these datasets have been acquired with different radiotracers, heterogeneous patient demographics, and various disease-specificities. In this work, we only consider the criteria of a single radiotracer when filtering PET images. For ADNI and OASIS-3, FBP and AV45 PET images were selected during filtering, respectively.

To facilitate our experiments, we divided each dataset randomly into five equal splits, each containing MRI-PET 3D paired samples. One split was allocated for testing, two splits for paired training data, and the remaining two splits for unpaired data. To ensure fairness, PET images from the last two splits (unpaired data) were paired only with MRI images from the extra unpaired MRI dataset, preventing any overlap between paired and unpaired images. This setup mirrors real-world application scenarios. This resulted in 1,088 paired and 1,088 unpaired samples in the ADNI training set, while the test set contained 543 paired samples. The OASIS training set comprised 308 paired and 308 unpaired samples, with 154 paired samples in the test set.

TABLE I: Evaluation results on the ADNI and OASIS datasets.

| Method | ADNI | | | | | | OASIS | | | | | |
| | MRI→PET | | | PET→MRI | | | MRI→PET | | | PET→MRI | | |
| | RMSE↓ | SSIM↑ | PSNR↑ | RMSE↓ | SSIM↑ | PSNR↑ | RMSE↓ | SSIM↑ | PSNR↑ | RMSE↓ | SSIM↑ | PSNR↑ |
|---|---|---|---|---|---|---|---|---|---|---|---|---|
| Pix2pix [12] | 0.079 | 0.914 | 27.589 | 0.331 | 0.342 | 15.665 | 0.091 | 0.915 | 26.747 | 0.191 | 0.630 | 20.446 |
| PAN [25] | 0.081 | 0.909 | 27.769 | 0.277 | 0.552 | 17.400 | 0.108 | 0.893 | 25.136 | 0.185 | 0.618 | 20.755 |
| CycleGAN [30] | 0.137 | 0.812 | 21.471 | 0.286 | 0.545 | 17.145 | 0.088 | 0.922 | 26.347 | 0.213 | 0.586 | 19.541 |
| PaPaGAN | **0.076** | **0.922** | **29.133** | **0.265** | **0.579** | **17.927** | **0.065** | **0.938** | **29.502** | **0.181** | **0.648** | **20.933** |

TABLE II: Significance test: p-values of paired t-test for methods mentioned in the first column rows (Pix2Pix, PAN, CycleGAN) w.r.t. PaPaGAN for ADNI and OASIS datasets.

| Dataset | ADNI | | | | OASIS | | | |
| | MRI → PET | | PET → MRI | | MRI → PET | | PET → MRI | |
| Method | SSIM | PSNR | SSIM | PSNR | SSIM | PSNR | SSIM | PSNR |
|---|---|---|---|---|---|---|---|---|
| Pix2Pix | 0.0432 | 7.21e-20 | 2.13e-209 | 2.57e-112 | 0.0005 | 1.08e-13 | 0.0251 | 7.24e-05 |
| PAN | 0.0060 | 1.55e-14 | 6.23e-05 | 1.57e-07 | 1.95e-11 | 4.23e-26 | 0.0028 | 0.0220 |
| CycleGAN | 4.49e-61 | 1.38e-150 | 4.21e-06 | 0.68e-07 | 1.55e-32 | 3.64e-14 | 1.61e-254 | 2.11e-318 |

*2) Data Preprocessing:* The T1-weighted MR data are rigidly aligned with an MNI template and then analyzed using FreeSurfer [7] (version 7) for intensity inhomogeneity correction and standardization followed by volumetric segmentation and cortical parcellation. PET imaging analysis was performed using the standard protocols that included scanner harmonization, motion correction, and PET-to-MR registration. We further processed the PET images through a unified pipeline [24] to extract regional values, culminating in the generation of intensity-normalized SUVR images. For these images, the cerebellum served as the reference region, providing a stable baseline for comparing and analyzing metabolic activity across the brain region. All images are processed to $256 \times 256 \times 256$.

*3) Implementation Details:* All methods utilized a 3D cubic input dimension of size $256 \times 256 \times 256$. To ensure regularization, a zero padding of 12 size was applied to each dimension, accompanied by RandomCrop. For paired data, the same cropping scheme was employed to maintain structural similarities between pairs. The training comprised 25,000 iterations with the Adam optimizer using $\beta$ values of (0.5, 0.999) and a batch size of 4. The learning rate was set to 2e-4 and underwent linear decay to zero after half the total iterations, with a warm-up period of 500 iterations initially. The same configuration was employed across all experiments (including baselines). The code accompanying this work will be released on the author's website.

We opt for a 3D-UNet [21] architecture with 3D convolution and 3D Instance Normalization operations and skip connections [10] to serve as our generator due to its efficient memory utilization. However, it's worth noting that any image translation framework could be utilized. The encoder of the UNet used a leaky relu activation, whereas the decoder used relu activation. The number of features increases from 12 to 192 in the encoder, and the decoder decreases it in reverse for unpaired generators. The paired generators are smaller and

contain features ranging from 8 to 64 only with a shallower depth. All discriminators adopt a 3D Patch discriminator design [12], enabling them to classify whether an image patch is real or synthetic. This choice of patch discriminators offers advantages such as parameter efficiency and improved regularization. It uses feature maps ranging from 16 to 128 based on the depth of the network. Each block uses a convolution layer followed by Instance Normalization and leaky relu activation except the first and the last block. The first block skips the normalization layer similar to [12], [30], and the last block consists of a single convolution that maps the features maps to a single dimension. The paired discriminator concatenates paired data along the channel dimension, whereas the unpaired discriminator uses the original input. The unpaired generator, paired generator, and each discriminator contain about 5.9 million, 1.8 million, and 1.75 million training parameters, respectively.

*B. Results and Analysis*

*1) Quantitative Results:* We employ Root Mean Squared Error (RMSE), Structural Similarity Index Measure (SSIM), and Peak Signal-to-Noise Ratio (PSNR) to assess the performance of various image translation models. We compared our approach against the following image translation methods: Pix2Pix, PAN, and CycleGAN, which belong to different settings of supervision. Pix2Pix and PAN are paired data-based image translation methods requiring full supervision. CycleGAN, on the other hand, is a popular unsupervised image-translation method that doesn't require any paired data. Xu et al. [27] is a semi-paired data-based framework similar to ours. However, we exclude this from the baseline comparison due to a lack of code availability and dataset differences. Tables I presents the quantitative outcomes of all methods across two directional image translation tasks: MRI→PET translation and PET→MRI translation. PaPaGAN consistently outperforms Pix2Pix, PAN, and CycleGAN across all metrics on all the

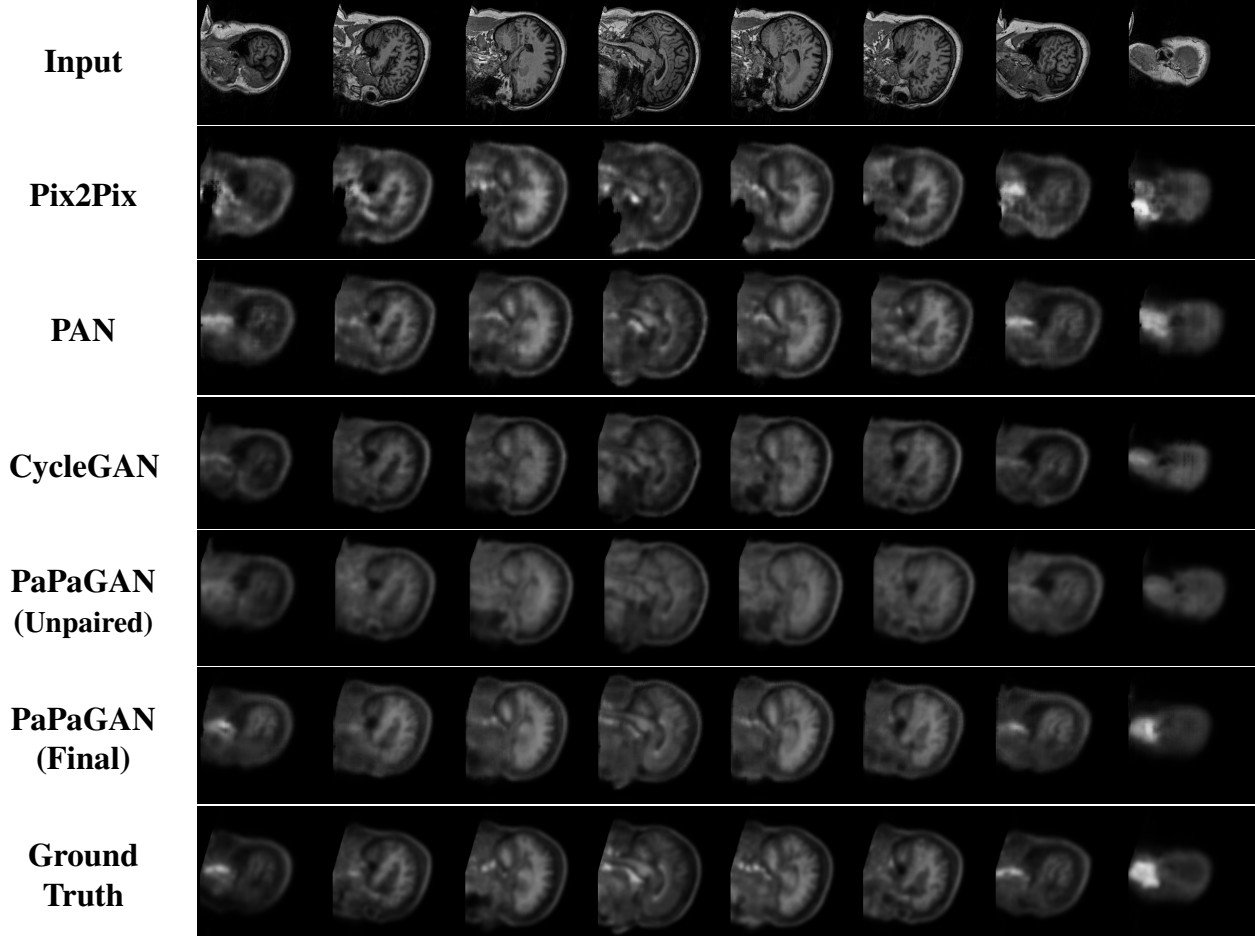

Fig. 2: Qualitative examples on MRI→PET translation from OASIS dataset. We show random crops of a few middle 2D central slices from 3D images. The first row shows the input MRI image to the model. The last row shows the ground truth, and the 2nd to 6th rows show the results of the different models. We see that PaPaGAN (Unpaired) missed some key aspects, but PaPaGAN (Final) fixed them and provided more accurate details than all the baselines.

tasks, underscoring its robustness. While Pix2Pix and PAN are confined to paired translation, relying solely on paired MRI-PET images, PaPaGAN capitalizes on both paired and unpaired MRI and PET images to enrich its image generator. CycleGAN, on the other hand, exhibits inferior results due to its reliance on unpaired data and the necessity to preserve unnecessary information for original image reconstruction. This leads to a degradation in its image translation quality.

Furthermore, our analysis reveals a notable discrepancy in translation performance between PET→MRI and MRI→PET directions. This discrepancy primarily stems from MRI images containing richer detail and high-frequency information, which aids the model in learning more precise MRI→PET translations. Conversely, the scarcity of detailed information in PET images poses challenges for generating high-quality MRI images in the reverse PET→MRI translation task. We believe that more training data should alleviate this issue.

To evaluate the statistical significance of our results, we conducted a paired t-test of all baseline methods with respect

to our framework, PaPaGAN. We set the following standards for evaluation:

- Null Hypothesis ($p > 0.05$): No significant difference between PSNR/SSIM of two methods
- Alternate Hypothesis ($p \leq 0.05$): The difference between PSNR/SSIM is significant (not by chance).

Table II demonstrates that for all the methods, PaPaGAN has significantly different quantitative metrics for both PSNR and SSIM in all cases. For all baselines compared in Table II, our framework has significantly low p-values, indicating that the quality of the translated images using our framework is higher.

*2) Output Visualization:* Figure 2 and 3 visualizes the results of our approach (PaPaGAN-Final) and baselines on the OASIS dataset for MRI→PET and PET→MRI using Sagittal view respectively. Evidently, the images generated by PaPaGAN exhibit superior visual quality and are closest to the ground truth. This indicates PaPaGAN's effective utilization of unpaired target domain images to achieve the desired appearance of target images. We also show the translated

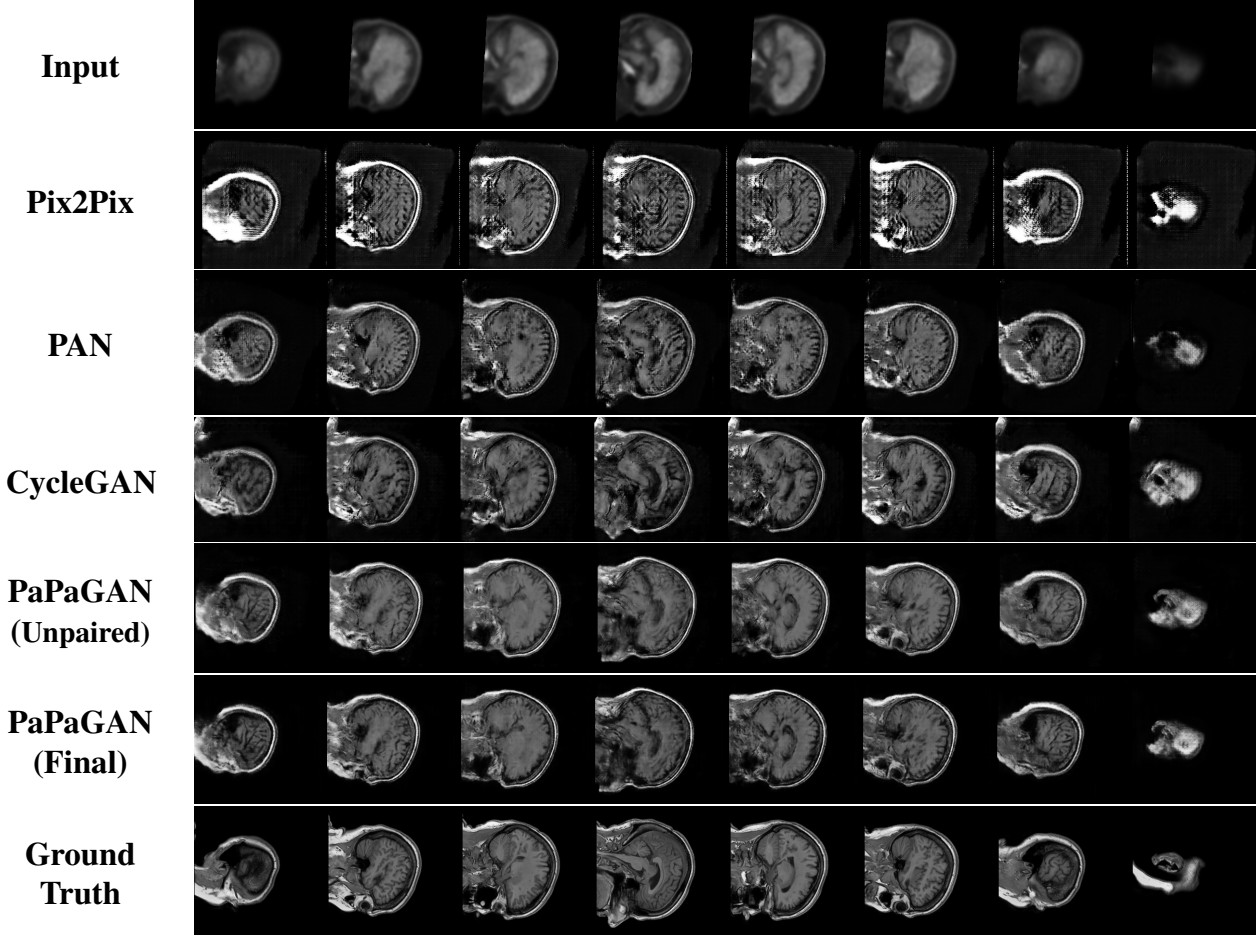

Fig. 3: Qualitative examples on PET→MRI translation from ADNI dataset.

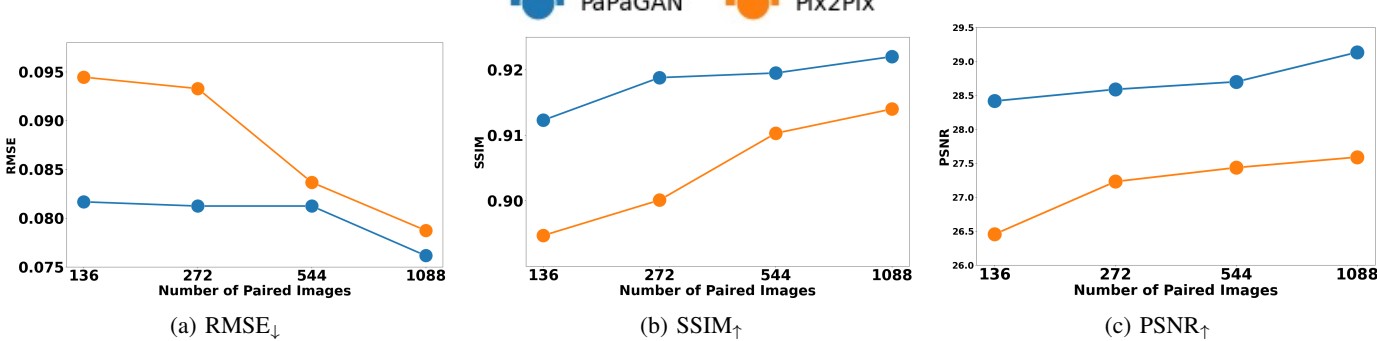

(a) RMSE↓            (b) SSIM↑            (c) PSNR↑

Fig. 4: Impact of the different number of Paired images from ADNI dataset (MRI → PET). We increased the number of unpaired images when decreasing paired images to keep the total number of images the same. We can observe that the unpaired images used in PaPaGAN help reduce RMSE and increase SSIM and PSNR.

output images of the unpaired generator in Figure 2 (PaPaGAN - Unpaired in the $5^{th}$ row). We can observe that the unpaired output is missing all the essential intensities, which are then corrected by the paired generator in the final output. The comparison in these figures highlights the significant advantages of PaPaGAN's approach. By integrating unpaired and paired data through a two-stage generation process, PaPaGAN can produce images that adhere closely to the desired target domain characteristics and retain essential details that might be lost when relying solely on unpaired data.

*3) Impact of the number of Paired Images:* We showcase the significance of paired and unpaired data in PaPaGAN by conducting experiments on the quantity of paired and unpaired images. Instead of employing two splits containing

paired images ($1,088$ paired images), we substitute paired data with unpaired data, i.e., we replace paired data with unpaired data. This keeps the total (paired+unpaired) the same but reduces paired data and increases unpaired data. Since Pix2Pix is trained exclusively on paired data, its training data is reduced, whereas PaPaGAN utilizes all the data. The findings of this experiment are illustrated in Figure 4. It shows that as the number of paired images decreases, the performance of pix2pix suffers significantly, whereas PaPaGAN exhibits a comparatively less decline in performance, highlighting the substantial impact of unpaired data.

## V. CONCLUSION

In this study, we introduce PaPaGAN, an innovative end-to-end medical image translation model leveraging GAN to utilize partially paired images. Our setting closely mimics real-world application scenarios, where a combination of paired and unpaired data is commonly encountered. PaPaGAN effectively utilizes both types of data to bolster the performance of the image translator by preserving the image content and providing high-quality outputs. Through comprehensive experiments across two-direction cross-modal image translation tasks and datasets, PaPaGAN demonstrates it outperformed the baselines in both quantitative and qualitative evaluations.

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
