# OpenReview forum: "Translation of Partially Paired Images with Generative Adversarial Networks"
_IEEE.org/EMBS/BHI/2024/Conference — IEEE BHI'24_

### Official Review · Reviewer_oK8g · 2024-08-08
**GAN framework designed to improve the translation of partially paired MRI-PET images**

**Overall Rating:** 6
**Confidence:** 2

**Other Quality Metrics:**

(a) Clarity of writing:good
(b) Clinical Significance:good
(c) Methodological Novelty: good
(d) Experiments and Results: fair

**Questions For The Authors:**

No

**Strengths:**

The two-stage translation process employed by PaPaGAN significantly enhances the quality and accuracy of the generated images and surpassed traditional methods that rely solely on either paired or unpaired data.

**Summary Of The Paper:**

The paper introduces PaPaGAN, a novel Generative Adversarial Network (GAN) designed to facilitate the translation of partially paired medical images, specifically between MRI (Magnetic Resonance Imaging) and PET (Positron Emission Tomography). This translation is valuable in clinical settings where paired MRI-PET data is often limited due to practical constraints such as cost and radiation exposure.The method demonstrates superior performance compared to existing paired and unpaired translation techniques, as validated by experiments on public datasets like ADNI and OASIS.

**Weaknesses:**

While PaPaGAN is designed to work with both paired and unpaired data, it would be critical to understand how sensitive its performance is to changes in the proportion of paired data available.The experiments primarily focus on MRI-PET translations using specific public datasets like ADNI and OASIS. It would be important to evaluate whether PaPaGAN can maintain its high performance across other imaging modalities or datasets with different characteristics, to assess its robustness and applicability in broader clinical contexts.

---

### Official Review · Reviewer_n2Te · 2024-08-10
**Translation of Partially Paired Images with Generative Adversarial Networks**

**Overall Rating:** 7
**Confidence:** 3

**Other Quality Metrics:**

(a) Clarity of writing;                  good
(b) Clinical Significance;            good
(c) Methodological Novelty;       good
(d) Experiments and Results      good

**Questions For The Authors:**

How can a clinician compare generated images to the ground truth if there isn't a paired image for reference?

**Strengths:**

The paper is well-written and easy to follow, with significant clinical utility for medical image translation.

**Summary Of The Paper:**

The paper presents PaPaGAN, an end-to-end Partially Paired Generative Adversarial Network designed specifically for partially paired images. Additionally, PaPaGAN utilizes unpaired data to learn a mapping function that generates a noisy intermediate image, enabling it to work effectively with both paired and unpaired data.

**Weaknesses:**

In the introduction, the paper claims that GANs are the state-of-the-art method for image generation, which contradicts the fact that Stable Diffusion has been the leading approach for this task. Additionally, the authors assert that their setting mimics real-world scenarios. However, it's unclear how this is feasible, as clinicians typically compare images to ground truth. How can they perform such comparisons if they don't have a paired image to reference?

---

### Official Review · Reviewer_7DVb · 2024-08-13
**The review of 'PaPaGAN: Translation of Partially Paired Images with Generative Adversarial Networks'**

**Overall Rating:** 7
**Confidence:** 4

**Other Quality Metrics:**

(a) Clarity of writing: Good
(b) Clinical Significance: Great
(c) Methodological Novelty: Great
(d) Experiments and Results: Good

**Questions For The Authors:**

Could a more systematic approach to model selection be provided? The ablation analysis in this study appears to be closely tied to hyperparameters like loss weight. I observed that the optimal number of unpaired samples saturates at a relatively high level (~544). A more thorough model selection analysis could potentially further enhance the impact of this work.

**Strengths:**

Impactful Topic: The translation between PET and MRI images is crucial for improving diagnostics and treatment in medical imaging. Addressing this challenge can significantly enhance patient care, particularly in neurodegenerative diseases.

Novelty in Model Design: PaPaGAN introduces a unique dual generator-discriminator architecture that leverages paired and unpaired data. This design overcomes the limitations of CycleGAN, enabling more accurate translations between domains with significant differences.

Comprehensive Evaluation: The model was rigorously tested against Pix2pix, PAN, and CycleGAN on the ADNI and OASIS datasets. PaPaGAN's superior performance in both PET-to-MRI and MRI-to-PET tasks underscores its robustness and practical relevance.

**Summary Of The Paper:**

The authors introduce PaPaGAN, a novel GAN-based model designed for translating between PET and MRI medical images. This approach employs dual generator-discriminator pairs, effectively leveraging both paired and unpaired image data. The proposed model addresses a key limitation observed in CycleGAN, which struggles to accurately translate features between domains with substantial differences.

PaPaGAN's performance was evaluated on the ADNI and OASIS datasets, where it demonstrated superior results compared to Pix2pix, PAN, and CycleGAN in both PET-to-MRI and MRI-to-PET translation tasks. These findings underscore the model's efficacy in utilizing partially paired data, leading to significant improvements in image translation accuracy.

**Weaknesses:**

The visualization for the more challenging PET-to-MRI tasks is notably absent, with no explanation provided. Additionally, the visualization lacks clarity regarding the biological significance of the generated images. For example, it is unclear whether the images accurately reflect cellular-level abnormalities or if a pathologist could reliably diagnose the disease based on these generated images.

---

### Decision · Program_Chairs · 2024-09-23

Accept